# Simultaneous Onset of Haematological Malignancy and COVID: An Epicovideha Survey

**DOI:** 10.3390/cancers14225530

**Published:** 2022-11-10

**Authors:** Chiara Cattaneo, Jon Salmanton-García, Francesco Marchesi, Shaimaa El-Ashwah, Federico Itri, Barbora Weinbergerová, Maria Gomes Da Silva, Michelina Dargenio, Julio Dávila-Valls, Sonia Martín-Pérez, Francesca Farina, Jaap Van Doesum, Toni Valković, Caroline Besson, Christian Bjørn Poulsen, Alberto López-García, Pavel Žák, Martin Schönlein, Klára Piukovics, Ozren Jaksic, Alba Cabirta, Natasha Ali, Uluhan Sili, Nicola Fracchiolla, Giulia Dragonetti, Tatjana Adžić-Vukičević, Monia Marchetti, Marina Machado, Andreas Glenthøj, Olimpia Finizio, Fatih Demirkan, Ola Blennow, Maria Chiara Tisi, Ali S. Omrani, Milan Navrátil, Zdeněk Ráčil, Jan Novák, Gabriele Magliano, Moraima Jiménez, Carolina Garcia-Vidal, Nurettin Erben, Maria Ilaria Del Principe, Caterina Buquicchio, Rui Bergantim, Josip Batinić, Murtadha Al-Khabori, Luisa Verga, Tomáš Szotkowski, Michail Samarkos, Irati Ormazabal-Vélez, Stef Meers, Johan Maertens, László Imre Pinczés, Martin Hoenigl, Ľuboš Drgoňa, Annarosa Cuccaro, Yavuz M. Bilgin, Avinash Aujayeb, Laman Rahimli, Stefanie Gräfe, Mariarita Sciumè, Miloš Mladenović, Gökçe Melis Çolak, Maria Vittoria Sacchi, Anna Nordlander, Caroline Berg Venemyr, Michaela Hanáková, Nicole García-Poutón, Ziad Emarah, Giovanni Paolo Maria Zambrotta, Raquel Nunes Rodrigues, Raul Cordoba, Gustavo-Adolfo Méndez, Monika M. Biernat, Oliver A. Cornely, Livio Pagano

**Affiliations:** 1Hematology Unit, ASST-Spedali Civili, 25123 Brescia, Italy; 2University of Cologne, Faculty of Medicine, University Hospital Cologne, Translational Research, Cologne Excellence Cluster on Cellular Stress Responses in Aging-Associated Diseases (CECAD), 50931 Cologne, Germany; 3University of Cologne, Faculty of Medicine, University Hospital Cologne, Department I of Internal Medicine, Center for Integrated Oncology Aachen Bonn Cologne Duesseldorf (CIO ABCD) and Excellence Center for Medical Mycology (ECMM), 50931 Cologne, Germany; 4Hematology and Stem Cell Transplant Unit, IRCCS Regina Elena National Cancer Institute, 00144 Rome, Italy; 5Oncology Center, Mansoura University, Mansoura 35516, Egypt; 6San Luigi Gonzaga Hospital, 10043 Orbassano, Italy; 7Masaryk University and University Hospital Brno—Department of Internal Medicine, Hematology and Oncology, 62500 Brno, Czech Republic; 8Portuguese Institute of Oncology, 1099-023 Lisbon, Portugal; 9Hematology and Stem Cell Transplan Unit, Vito Fazzi, 73100 Lecce, Italy; 10Hospital Nuestra Señora de Sonsoles, 05004 Ávila, Spain; 11IRCCS Ospedale San Raffaele, 20132 Milan, Italy; 12University Medical Center Groningen, 9713 GZ Groningen, The Netherlands; 13University Hospital Centre Rijeka, 51000 Rijeka, Croatia; 14Croatian Cooperative Group for Hematological Diseases (CROHEM), 10000 Zagreb, Croatia; 15Faculty of Medicine, University of Rijeka, 51000 Rijeka, Croatia; 16Centre Hospitalier de Versailles, Le Chesnay, Université Paris-Saclay, UVSQ, Inserm, Équipe “Exposome et Hérédité”, CESP, 94800 Villejuif, France; 17Zealand University Hospital, 4000 Roskilde, Denmark; 18Fundacion Jimenez Diaz University Hospital, Health Research Institute IIS-FJD, 28040 Madrid, Spain; 19University Hospital Hradec Králové, 50005 Hradec Králové, Czech Republic; 20Department of Oncology, Hematology and Bone Marrow Transplantation with Section of Pneumology, University Medical Center Hamburg-Eppendorf, 20251 Hamburg, Germany; 21Department of Internal Medicine, South Division, Faculty of Medicine University of Szeged, 6720 Szeged, Hungary; 22University Hospital Dubrava, 10000 Zagreb, Croatia; 23Department of Hematology, Vall d’Hebron Hospital Universitari, Experimental Hematology, Vall d’Hebron Institute of Oncology (VHIO), Vall d’Hebron Barcelona Hospital Campus, 08193 Barcelona, Spain; 24Departament de Medicina, Universitat Autònoma de Barcelona, 08193 Bellaterra, Spain; 25Aga Khan University, 74000 Karachi, Pakistan; 26Department of Infectious Diseases and Clinical Microbiology, School of Medicine, Marmara University, 34854 Istanbul, Turkey; 27Hematology Unit, Fondazione IRCCS Ca’ Granda Ospedale Maggiore Policlinico, 20122 Milan, Italy; 28Hematology Unit, Fondazione Policlinico Universitario Agostino Gemelli—IRCCS, 00168 Rome, Italy; 29COVID Hospital “Batajnica”, 11000 Belgrade, Serbia; 30Azienda Ospedaliera Nazionale SS. Antonio e Biagio e Cesare Arrigo, 15121 Alessandria, Italy; 31Clinical Microbiology and Infectious Diseases Department, Hospital General Universitario Gregorio Marañón, 28009 Madrid, Spain; 32Department of Hematology, Copenhagen University Hospital—Rigshospitalet, DK-2100 Copenhagen, Denmark; 33UOC Hematology, AORN Cardarelli, 80131 Naples, Italy; 34Dokuz Eylul University, Division of Hematology, 35340 Izmir, Turkey; 35Department of Infectious Diseases, Karolinska University Hospital, 17176 Stockholm, Sweden; 36Ospedale San Bortolo, 36100 Vicenza, Italy; 37Communicable Disease Center, Hamad Medical Corporation, Doha 3050, Qatar; 38University Hospital Ostrava, 70852 Ostrava, Czech Republic; 39Institute of Hematology and Blood Transfusion, 12800 Prague, Czech Republic; 40Department of Hematology, University Hospital Kralovske Vinohrady and Third Faculty of Medicine, Charles University, 12808 Prague, Czech Republic; 41ASST Grande Ospedale Metropolitano Niguarda, 20162 Milan, Italy; 42Department of Hematology, University Hospital Vall d’Hebron, Experimental Hematology, 08035 Barcelona, Spain; 43Vall d’Hebron Institute of Oncology (VHIO), Vall d’Hebron Barcelona Hospital Campus, 08035 Barcelona, Spain; 44Department of Infectious Diseases, Hospital Clinic de Barcelona, University of Barcelona, IDIBAPS, 08035 Barcelona, Spain; 45Department of Infectious Diseases and Clinical Microbiology, Faculty of Medicine Eskisehir Osmangazi University, 26040 Eskisehir, Turkey; 46Hematology, Department of Biomedicine and Prevention, University of Rome Tor Vergata, 00133 Rome, Italy; 47Ematologia Con Trapianto, Ospedale Dimiccoli Barletta, 70051 Barletta, Italy; 48Centro Hospitalar e Universitário São João, 4200-319 Porto, Portugal; 49University Hospital Centre Zagreb, 10000 Zagreb, Croatia; 50Croatian Cooperative Group for Hematological Diseases (KROHEM), 10000 Zagreb, Croatia; 51School of Medicine University of Zagreb, 10000 Zagreb, Croatia; 52Sultan Qaboos University Hospital, Muscat 123, Oman; 53Azienda Ospedaliera San Gerardo—Monza, 20900 Monza, Italy; 54Università Milano-Bicocca, 20126 Milan, Italy; 55University Hospital Olomouc, 77900 Olomouc, Czech Republic; 56Laikon Hospital, 11527 Athens, Greece; 57Hospital Universitario de Navarra, 31008 Iruña-Pamplona, Spain; 58AZ KLINA, 2930 Brasschaat, Belgium; 59KU Leuven, 3000 Leuven, Belgium; 60Division of Hematology, Department of Internal Medicine, Faculty of Medicine, University of Debrecen, H-4032 Debrecen, Hungary; 61Division of Infectious Diseases and Global Public Health, Department of Medicine, University of California San Diego, San Diego, CA 92093, USA; 62Clinical and Translational Fungal-Working Group, University of California San Diego, La Jolla, CA 92093, USA; 63Division of Infectious Diseases, Department of Internal Medicine, Medical University of Graz, A-8036 Graz, Austria; 64Department of Oncohematology, Comenius University and National Cancer Institute, 81499 Bratislava, Slovakia; 65Hematology Unit, Center for Translational Medicine, Azienda USL Toscana NordOvest, 55100 Livorno, Italy; 66Department of Internal Medicine, ADRZ, 4462 RA Goes, The Netherlands; 67Northumbria Healthcare, Newcastle NE29 8NH, UK; 68Universitätsklinikum Hamburg Eppendorf, 20251 Hamburg, Germany; 69Clinic for Orthopedic Surgery and Traumatology, University Clinical Center of Serbia, 11000 Belgrade, Serbia; 70Hospital Escuela de Agudos Dr. Ramón Madariaga, Posadas 3736, Argentina; 71Department of Haematology, Blood Neoplasms, and Bone Marrow Transplantation, Wroclaw Medical University, 50-425 Wroclaw, Poland; 72University of Cologne, Faculty of Medicine, University Hospital Cologne, Clinical Trials Centre Cologne (ZKS Köln), 50931 Cologne, Germany; 73German Centre for Infection Research (DZIF), Partner Site Bonn-Cologne, 50931 Cologne, Germany; 74Hematology Unit, Università Cattolica del Sacro Cuore, 86100 Rome, Italy

**Keywords:** haematological malignancy onset, COVID-19, treatment, outcome, prognostic factors

## Abstract

**Simple Summary:**

Patients with simultaneous diagnosis of haematological malignancies (HM) and COVID-19 are an even greater challenge for hematologists. To better clarify their outcome, we describe the clinical features and outcome of a cohort of 450 patients with simultaneous diagnosis of HM and COVID-19 registered in the EPICOVIDEHA registry between March 2020 to February 2022. Overall, 343 (76.2%) patients received treatment for HM, and an overall response rate was observed in 140 (40.8%) patients after the first line of treatment. Thirty-day mortality was significantly higher in patients not receiving HM treatment (42.1%) than in those receiving treatment (27.4%, *p* = 0.004). Statistical analysis showed that, together with age, severe/critical COVID-19, ≥2 comorbidities, lack of HM treatment was an independent risk factors for mortality. These observations suggest the importance of HM treatment in these patients; therefore, it should be delivered as soon as possible for patients requiring immediate therapy.

**Abstract:**

Background: The outcome of patients with simultaneous diagnosis of haematological malignancies (HM) and COVID-19 is unknown and there are no specific treatment guidelines. Methods: We describe the clinical features and outcome of a cohort of 450 patients with simultaneous diagnosis of HM and COVID-19 registered in the EPICOVIDEHA registry between March 2020 to February 2022. Results: Acute leukaemia and lymphoma were the most frequent HM (35.8% and 35.1%, respectively). Overall, 343 (76.2%) patients received treatment for HM, which was delayed for longer than one month since diagnosis in 57 (16.6%). An overall response rate was observed in 140 (40.8%) patients after the first line of treatment. After a median follow-up of 35 days, overall mortality was 177/450 (39.3%); 30-day mortality was significantly higher in patients not receiving HM treatment (42.1%) than in those receiving treatment (27.4%, *p* = 0.004), either before and/or after COVID-19, or compared to patients receiving HM treatment at least after COVID-19 (15.2%, *p* < 0.001). Age, severe/critical COVID-19, ≥2 comorbidities, and lack of HM treatment were independent risk factors for mortality, whereas a lymphocyte count >500/mcl at COVID-19 onset was protective. Conclusions: HM treatment should be delivered as soon as possible for patients with simultaneous diagnosis of COVID-19 and HM requiring immediate therapy.

## 1. Introduction

The severe acute respiratory syndrome coronavirus 2 (SARS-CoV-2) was declared a global pandemic by the World Health Organization (WHO) in March 2020. Since then, many reports demonstrated high morbidity and mortality in haematological patients with coronavirus disease 2019 (COVID-19), partly driven by uncontrolled haematological malignancy (HM), which can affect the severity of COVID-19 [1,2,3].

However, to date, few data are available about the impact of starting early versus delayed specific HM treatment as well as the outcome of patients with simultaneous onset of HM and COVID-19. Some recommendations from expert panels have raised concern regarding the management of specific HM, including acute leukaemia (AL), multiple myeloma (MM) and hairy cell leukaemia (HCL), during the pandemic [4,5,6,7,8]. In these recommendations, few suggestions apply to patients with simultaneous HM and COVID-19 diagnosis, and the advice is usually to delay, when possible, the start of HM therapy. Indeed, the severity of the COVID-19 clinical picture might make any HM specific therapeutic approach impossible, but some cases of simultaneous diagnosis of COVID-19 and AL requiring immediate treatment have been reported, suggesting that the HM therapeutic approach may be possible even with an active infection [9,10]. Other researchers have indicated that an intensity-reduced approach may be helpful, particularly in the management of specific HM, such as acute myeloid leukaemia (AML) [11].

Specific data on a large cohort of patients with simultaneous onset of HM and COVID-19 with long-term follow-up are still lacking, and there are no evidence-based algorithms to guide clinicians in the selection of the best therapeutic approach and timing, particularly for patients requiring urgent antineoplastic treatment.

Hereby, we aimed to assess the overall survival (OS) of patients with simultaneous onset of HM and COVID-19 as primary endpoint of the study. In addition, we intended to determine: (1) the HM response rate of patients with simultaneous onset of HM and COVID-19; (2) the rate of patients receiving specific treatment for HM during the acute phase of COVID-19; (3) the prognostic factors in patients with simultaneous onset of HM and COVID-19; (4) the time interval between HM diagnosis and initiation of therapy; (5) the safety of specific haematological treatments during COVID-19.

## 2. Materials and Methods

We have conducted an observational multicentre analysis of patients with HM who developed COVID-19 between March 2020 and February 2022. Data were collected from the EPICOVIDEHA registry. EPICOVIDEHA (www.clinicaltrials.gov; NCT04733729) is an international open web-based registry for patients with HM infected with SARS-CoV-2. This registry was centrally approved by the local ethics committee of the Fondazione Policlinico Universitario Agostino Gemelli—IRCCS, Università Cattolica del Sacro Cuore of Rome, Italy (Study ID: 3226). Additionally, if applicable, the respective local ethics committee of each participating institution might have approved the EPICOVIDEHA. EPICOVIDEHA’s methods have been described elsewhere [12]. The electronic case report form (eCRF) is accessible online at www.clinicalsurveys.net (EFS Summer 2021, TIVIAN, Cologne, Germany, Accessed on 23 February 2022). Each documented patient was reviewed and validated by experts in infectious diseases and haematology. Inclusion criteria were described as follows: (a) new diagnosis of HM within 33 days before or 3 days after SARS-CoV-2 infection diagnosis, (b) adult patients ≥ 18 years old, and (c) laboratory-based diagnosis of SARS-CoV-2 infection.

Data on patient baseline conditions pre-COVID-19 (i.e., age, sex, factors predisposing for COVID-19, COVID-19 vaccination status), HM clinical management (i.e., type of treatment received and any delay), COVID-19 severity, and outcome (i.e., mortality, attributable mortality [assessed by the medical team in charge of the patient], last day of follow-up) were collected. COVID-19 severity was graded according to international standards as previously described (asymptomatic: no clinical signs or symptoms; mild: non-pneumonia and mild pneumonia; severe: dyspnoea, respiratory frequency ≥ 30 breaths per min, SpO_2_ ≤ 93%, PaO_2_/FiO_2_ < 300, or lung infiltrates > 50%); critical: patients admitted to intensive care for respiratory failure, septic shock, or multiple organ dysfunction or failure) [1] HM treatment was considered delayed if delivered more than one month after the HM diagnosis. Overall response rate for HM included complete remission and partial remission [13]. Neutropenia and lymphopenia were reported based on grade 3 and 4 values according to Common Terminology Criteria for Adverse Events (CTCAE) v6.0 [14]. The observation period was split into four intervals (1st: March 2020–September 2020; 2nd: October 2020–February 2021; 3rd: March 2021–November 2021; 4th: December 2021–February 2022), according to pandemic waves and to vaccine availability.

No a priori sample size calculation was performed for this analysis. Categorical variables are described using frequencies and percentages, whereas continuous variables using median, interquartile range (IQR) and absolute range. A Cox regression hazard model was designed and run with variables suspected to play a role in the mortality of patients with COVID-19, as previously described [11]. A multivariable Cox regression model was calculated with the Wald backward method, including only the variables statistically significant in the univariable model. Mortality was analysed using Kaplan–Meier survival plots. Log-rank test was used to compare the survival probability of the patients included in the different plots. A *p*-value ≤ 0.05 was found statistically significant. SPSSv27.0 was employed for statistical analyses (SPSS, IBM Corp., Chicago, IL, USA).

## 3. Results

### 3.1. Characteristics of Patients

From March 2020 to February 2022, 450 patients with simultaneous onset of HM and COVID-19 have been reported in the EPICOVIDEHA registry. Of these, 118 (26.2%) patients were diagnosed with COVID-19 between March and September 2020, 167 (37.1%) between October 2020 and February 2021, 71 (15.8%) between March and November 2021, and 94 (20.9%) between December 2021 and February 2022.

Patient characteristics are summarized in Table 1. Males were predominant (n = 264, 58.7%), with a median age of 65 years (IQR: 53 to 68). At least one comorbidity was reported in 140 (62.7%) patients, and two in 142 (31.6%). AL, particularly AML, were the most common HM (AL n = 161, 35.8%; AML n = 129, 28.7%), together with non-Hodgkin Lymphoma (NHL), mainly aggressive (NHL n = 142, 31.6%; aggressive NHL n = 91, 20.2%). The median time between HM and COVID-19 diagnosis was 11 days (IQR: −22 to −2) before COVID-19. At COVID-19 diagnosis, neutropenia was present in 97 (21.5%) patients (grade 4 in n = 69, 15.5% with less than 500 neutrophils/mm^3^), and lymphopenia in 101 (22.4%) patients. Secondary infections were reported in 63 (14%) of patients; of these, 6 (9.5%) were invasive possible or probable mould diseases. One fifth of the patients (n = 89, 19.8%) received at least one vaccine dose before COVID-19 diagnosis, with 29 (6.4%) being fully vaccinated; they were recorded all in the 3rd and 4th intervals. In 70 cases patients were vaccinated with mRNA based vaccines. COVID-19 was severe or critical in 286 (63.6%) patients. Most of the patients were admitted to hospital (n = 380, 84.4%); a specific treatment for COVID-19 was delivered in 173 (38.4%) patients, as reported in Table 1. In Appendix A COVID-19 directed antivirals and monoclonal antibodies are reported.

### 3.2. HM Treatment

Overall, at least 343 (76.2%) patients with simultaneous COVID-19 and HM received treatment for HM, either before and/or after COVID-19 diagnosis. HM treatment was delivered at least before COVID-19 diagnosis in 227/343 (66.2%) patients, with a median time between HM diagnosis and start of HM treatment of 7 days (IQR: 3 to 19). HM treatment was started only after COVID-19 in 116 (33.8%) patients. Only 41 (5.6%) patients were treated from day 32 of COVID-19 onwards. Forty-five (10.0%) patients were treated only with palliative care, before or after COVID-19. One-hundred and seven (23.7%) patients did not receive any treatment until the last day of follow-up. In seven (1.5%) cases no information about HM treatment was provided.

HM treatment was delayed beyond one month since HM diagnosis in 57 (17.2%) patients. Treatment delay was rarer for AL (AML: 9/103, 8.7%, acute lymphoblastic leukaemia, ALL: 2/27, 9.1%) and MM (5/41, 12.2%) and more frequent in NHL (27/112, 21.7%), both aggressive (21.9%) and indolent (20.8%), myelodysplastic/myeloproliferative syndromes (MDS/MPS) (5/22, 22.7%), chronic lymphoproliferative disorders (CLD) (4/11, 36.4%) and Hodgkin lymphoma (HL) (7/14, 50%). In Appendix A all the treatments for HM delivered before and after COVID-19 diagnosis are detailed.

Table 2 summarizes the clinical characteristics of patients by HM treatment. Patients receiving HM treatment (either before or after COVID-19) were more often female. AML or lymphoma patients were more frequently treated than those with CLD or MDS/MPS. The number of comorbidities and COVID-19 severity, as well as vaccination status and specific treatment for COVID-19 did not differ significantly in treated and untreated patients. The largest number of HM treated patients was observed in the second period (October 2020–February 2021). Thirty-day mortality was significantly lower in treated patients for HM as compared to those receiving no HM treatment.

### 3.3. Patient Outcome

Among the 343 treated patients, HM treatment overall response rate was 40.8% (140/343 patients) after the first line of treatment; 100 (29.2%) patients had complete remission. Chemotherapy-induced neutropenia occurred in 155 (45.2%) patients and secondary infections were reported in 85 (24.8%).

After a median follow-up of 35 days (IQR: 12-168) since COVID-19 diagnosis, 177/450 (39.3%) had died. Thirty-day mortality was 139/450 (30.9%) for the whole cohort. It was higher in AML (57/129, 44.2%) and in MM patients (21/55, 38.2%) and lower in lymphoma (39/156, 25%), in MDS/MPS (9/39, 23.1%), in ALL (6/32, 18.7%) and in CLD (7/39, 17.9%) patients (overall survival probability reported in Figure 1a,b). Thirty-day mortality was similar in indolent and aggressive NHL (respectively: 12/42, 28.6% and 26/91, 28.6%), regardless of different proportions of HM treatment received (24/42, 57.1% and 81/91, 90.1%, respectively). In 151 (85.3%) patients COVID-19 was responsible for death; however, in 78 (44%) patients the main reason for death was COVID-19, in 73 (41.2%) COVID-19 and HM and in 26 (5.8%) HM. Thirty-day mortality was significantly higher in patients not receiving HM treatment (42.1%) than in those receiving treatment (27.4%, *p* = 0.004), either before and/or after COVID-19, or compared to patients receiving HM treatment at least after COVID-19 (15.2%, *p* < 0.001). The lowest mortality (12.1%) was observed in patients receiving treatment only after COVID-19. Mortality rates by timing of HM treatment are showed in Appendix A.

Overall, patients treated for HM showed a higher survival probability at 30 days follow-up as compared to those untreated (*p* < 0.001) (Figure 2). Higher survival probability in those who were treated was also found in patients with AML (*p* < 0.001), lymphoma (*p* < 0.001), MDS/MPS (*p* = 0.01), and MM (*p* = 0.052) (Appendix A).

### 3.4. Risk Factors for Mortality

Increasing age, AML diagnosis, pre-existing comorbidities, critical COVID-19, and no HM treatment were observed as risk factors for mortality in the univariable analysis, whereas simultaneous HM and COVID-19 diagnosis in the 2nd and 4th period, a lymphocyte count > 500/mcl at COVID-19 onset, administration of at least conventional chemotherapy, administration of at least immunotherapy before COVID-19, HM treatment after COVID-19, and anti-SARS-CoV-2 monoclonal antibody administration were protective. At multivariable analysis, age, report of at least two comorbidities, severe/critical COVID-19, and no treatment were independently associated with an increased mortality, whereas a lymphocyte count >500/mcl at COVID-19 onset was protective. Chemotherapy induced neutropenia during the acute phase of COVID-19 did not show any impact on mortality (Table 3).

Age, critical COVID-19, and no HM treatment were risk factors for mortality at univariable and multivariable analyses for both AML and lymphoma patients. Chemotherapy induced neutropenia during the acute phase of COVID-19 was not a mortality risk factor in AML cases, whereas it was for lymphoma patients; immunotherapy before COVID-19 was protective at univariable analysis for lymphoma patients. Moreover, the diagnosis of COVID-19 in consecutive waves to the initial one was associated with a reduced risk for mortality as compared to the first in lymphoma patients at multivariable analysis, while this was not significant for AML (Appendix A).

## 4. Discussion

The COVID-19 pandemic has impacted the algorithms for treating HM. While there is some consensus on when to avoid or delay HM treatment during the pandemic [4], no guidelines are available yet for treatment of patients with simultaneous COVID-19 and HM diagnosis. Here we present results of a dataset of 450 patients with simultaneous diagnosis of HM and COVID-19, derived from the EPICOVIDEHA registry. In this analysis, AL and lymphoma were predominant, and about three quarters of these patients received treatment for HM. An overall response rate was observed in about 40% of patients after first-line treatment, and overall survival was 60.7%. Age, critical COVID-19, ≥2 comorbidities and lack of HM treatment were independent predictors of mortality.

The demographic characteristics of patients were similar to those observed in the overall population of the EPICOVIDEHA registry [1] and in other series [2,3]. Most patients were admitted to hospital, with a very high proportion of AL patients, particularly AML, which could be justified by the clinical complexity of these patients who cannot be managed at home since the onset of HM. Indeed, more than half of patients showed a severe or critical COVID-19, which is the subset more difficult to treat for the haematologists.

Among patients receiving HM treatment, 66.2% initiated their treatment before COVID-19 diagnosis and only 33.8% after. As compared to non-treated patients, treated patients were more often female and younger, which could be explained by a relative lower severity of COVID-19. Treated patients were more often affected by AL and, although to a lesser extent, lymphoma, probably because of the urgent need of treatment for these patients. Severity of COVID-19 did not differ significantly in treated and untreated patients, whereas the degree of severity was less pronounced when considering patients treated at least after COVID-19. Thirty-day mortality was significantly lower in patients treated either before and/or after COVID-19 or only after COVID-19 compared to patients with no treatment at all, suggesting that early HM treatment contributes to a better control of SARS-CoV-2 infection.

Overall, the response rate to HM treatment was relatively low (40.8%), probably also for the proportion of deceased patients during treatment, and after a median follow-up of 35 days 60.7% of patients remained alive. COVID-19 alone or in association with HM was the main cause of death. As expected, AML was the HM with the highest 30-day mortality (44.2%), confirming data already reported in the pre-vaccination era [1,5,15]. Mortality was markedly lower (18.7%) in ALL. Causes of this difference may be found in the usually younger age of ALL patients as compared to AML and in the less intensive induction regimen required for ALL treatment than AML, in addition to the fact that also steroid alone can control the disease, at least for a limited period. A Spanish study [16] reported a mortality of 29% in an ALL cohort including patients on salvage treatment during the first two COVID-19-waves. A lower mortality (11.1%) in ALL has also been reported by Chiaretti et al. [17] in a series of 63 ALL COVID-19 patients in a recent observational study, where only a minority (22.2%) of patients became infected with SARS-CoV-2 at diagnosis.

Considering lymphoproliferative disorders, a wide range of mortality rates has been observed among MM, lymphoma and CLD patients in previous EPICOVIDEHA studies [1], with the worst prognosis for MM. As reported in other series enrolling patients during different phases of HM [2,3], it is interesting to notice that no difference in mortality has been recorded between aggressive and indolent NHL COVID-19 patients neither at HM onset, probably reflecting an intrinsic immune response impairment of NHL patients.

Lack of HM treatment was an independent risk factor for mortality, in addition to well-established risks such as age, COVID-19 severity and presence of comorbidities. This association was quite evident in some HM patients, particularly AML, lymphoma, MDS/MPS and MM; it highlights the crucial role of early HM treatment and, therefore, of disease control, for a favourable outcome. In our series, chemotherapy induced neutropenia during the acute phases of COVID-19 did not affect the outcome of AML patients, as opposed to lymphoma patients, where it did. The role of neutropenia as negative impact on AML patients is still a matter of debate. Although some series report a negative impact of neutropenia [18,19,20,21], in the AML EPICOVIDEHA study [5] neutropenia at onset of COVID-19 was not a risk factor for mortality. Indeed, neutropenia is an intrinsic feature of AML, and it is present in AML patients regardless of treatment received. This is not true for lymphoma patients, where chemotherapy induced neutropenia had a negative impact on survival, confirming that less intensive treatment should be preferred during the acute phase of COVID-19, as also suggested by the recent ECIL recommendations [22]. The incidence of secondary infections was relatively low (14%), and invasive fungal diseases were even rarer. These findings are in line with other reports on ICU patients [23,24] or HM patients [25].

In our cohort, less than ten percent of patients were fully vaccinated against COVID-19. Therefore, our study was not able to appropriately evaluate the impact of vaccination on outcome. We also analysed the outcome according to the pandemic period and we observed a relationship only for lymphoma patients, where any period after the first was independently associated to a better outcome. The mortality risk reduction across pandemic periods could be explained by multiple factors, such as an increased number of asymptomatic or mild infections registered while the pandemic was progressing and maybe also a lower virulence of SARS-CoV-2 infection detected over time [26], while in this study COVID-19 specific treatments did not have a relevant impact. Interestingly, in contrast to some of the largest epidemiologic studies published so far [1,5], no impact of pandemic period was observed for AML and simultaneous COVID-19 diagnosis patients, probably because of the burden of active HM disease.

The study is subject to several limitations. First, the absence of a comparison group of HM patients without COVID-19 makes it difficult to assess the real impact of covid on the prognosis of haematological disease and vice versa. Moreover, the choice of the time frame (−33 days to +3 days from COVID-19) for the simultaneous diagnosis of HM and COVID-19 was chosen based on the enrolment criteria of the protocol, which planned to record only patients who had a history of HM. We decided to not go beyond 33 days before COVID-19 to avoid enrolling patients who have already undergone more than one course of HM treatment. Another limitation of the study is the heterogeneity of the HM population and the different algorithms of HM treatment adopted in different countries during the COVID-19 pandemic, which makes interpretation of the results more difficult. Indeed, the concept of treatment delay is not the same for aggressive and indolent HM diseases. Another limitation is that we do not know exactly how many patients went from curative intent to palliation after COVID-19. Moreover, the inclusion of patients in the registry was potentially influenced by the impact of the pandemic waves, and results may not be applicable to the current omicron wave. Finally, the low percentage of vaccinated cases as well as those receiving COVID-19 treatment with monoclonal antibodies limit the applicability to the current and potentially also future phases of the pandemic.

## 5. Conclusions

Patients with simultaneous COVID-19 and HM still represent a challenge for haematologists, particularly in elderly patients with AML. Our results outline the importance of HM treatment delivered as soon as possible for patients requiring immediate HM therapy. Indeed, Haematologists may fear initiating treatment that may cause more immunosuppression or lead to other side effects in a patient with active COVID-19; however, the consequences of delayed or suspended treatment may be worse. Any supportive measure should be provided for those patients with chemotherapy-induced neutropenia. Whenever possible, a less-intensive approach should be considered, particularly for lymphoma patients, where chemotherapy-induced neutropenia was found as risk factor for mortality. Future studies should evaluate the impact of large-scale COVID-19 vaccination on outcome of patients with simultaneous HM and COVID-19 diagnosis.

## Figures and Tables

**Figure 1 cancers-14-05530-f001:**
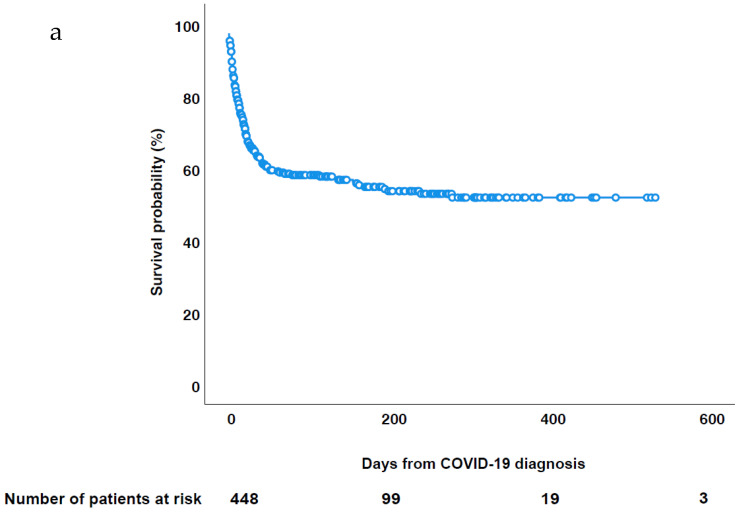
Overall survival probability. (**a**) Cohort. (**b**) Patients by malignancy.

**Figure 2 cancers-14-05530-f002:**
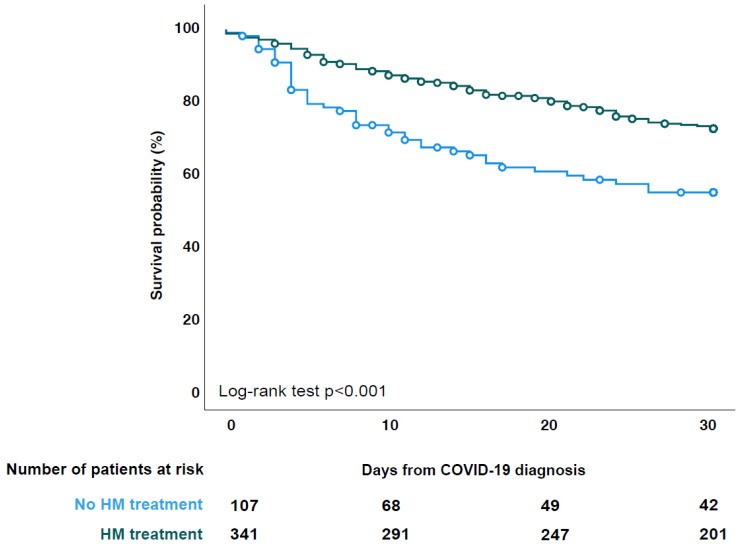
30-day survival probability of treated vs. untreated patients.

**Table 1 cancers-14-05530-t001:** Characteristics of patients with simultaneous onset of haematological malignancy and COVID-19.

	n	%
Sex		
Female	186	41.3%
Male	264	58.7%
Age, median (IQR) [absolute range]	65 (53 to 75) [18 to 95]	
18–25 years old	15	3.3%
26–50 years old	84	18.7%
51–69 years old	174	38.7%
≥70 years old	177	39.3%
Comorbidities before COVID-19		
No comorbidities	168	37.3%
1 comorbidity	140	31.1%
2 comorbidities	81	18%
3 or more comorbidities	61	13.6%
Chronic cardiopathy	156	34.7%
Chronic pulmonary disease	61	13.6%
Diabetes mellitus	77	17.1%
Liver disease	18	4%
Obesity	43	9.6%
Renal impairment	43	9.6%
Smoking history	52	11.6%
No risk factor identified	168	37.3%
Baseline malignancy		
Days between malignancy diagnosis and COVID-19 diagnosis, median (IQR) [absolute range]	−11 (−21 to −2) [−33 to 3]	
Acute lymphoid leukaemia	32	7.1%
Chronic lymphoid leukaemia	27	6%
Acute myeloid leukaemia	129	28.7%
Chronic myeloid leukaemia	9	2%
Hodgkin lymphoma	16	3.6%
Non-Hodgkin lymphoma	142	31.6%
Indolent	42	9.3%
Aggressive	91	20.2%
Unknown	7	1.6%
Essential thrombocythemia	2	0.4%
Multiple myeloma	55	12.2%
Myelodysplastic syndrome	25	5.6%
Myelofibrosis	3	0.7%
Hairy cell leukaemia	10	2.2%
Neutrophils at COVID-19 diagnosis		
≤500	69	15.3%
501–999	28	6.2%
≥1000	307	68.2%
Lymphocytes at COVID-19 diagnosis		
≤200	47	10.4%
201–499	54	12%
≥500	315	70%
COVID-19 infection		
Asymptomatic	106	23.6%
Mild infection	58	12.9%
Severe infection	170	37.8%
Critical infection	116	25.8%
COVID-19 symptoms at infection onset		
Pulmonary	151	33.6%
Pulmonary + extrapulmonary	106	23.6%
Extrapulmonary	66	14.7%
Screening	127	28.2%
COVID-19 vaccination		
Not vaccinated	360	80%
One dose	15	3.3%
Two doses	46	10.2%
Three doses	29	6.4%
COVID-19 treatment		
No specific treatment reported	277	61.6%
Antivirals + monoclonal antibodies ± corticosteroids ± plasma	8	1.8%
Antivirals ± corticosteroids ± plasma	52	11.6%
Corticosteroids	72	16%
Monoclonal antibodies ± plasma ± corticosteroids	31	6.9%
Plasma ± corticosteroids	10	2.2%
Stay during COVID-19 episode		
Hospital	380	84.4%
Days of the stay in ICU, median (IQR) [absolute range]	15 (7 to 28) [1 to 150]	
ICU	116	25.8%
Days of the stay in ICU, median (IQR) [absolute range]	8 (4 to 20) [1 to 56]	
Invasive mechanical ventilation	74	16.4%
Non-invasive mechanical ventilation	42	9.3%
Home	93	20.7%

Negative days indicate days before COVID-19 diagnosis. Positive days indicate days after COVID-19 diagnosis. Day of COVID-19 diagnosis is set as day 0. Asymptomatic: no clinical signs or symptoms; mild: non-pneumonia and mild pneumonia; severe: dyspnea, respiratory frequency ≥ 30 breaths per min, SpO_2_ ≤ 93%, PaO_2_/FiO_2_ < 300, or lung infiltrates > 50%); critical: patients admitted in intensive care for respiratory failure, septic shock, or multiple organ dysfunction or failure. COVID-19, coronavirus disease 2019; ICU, intensive care unit; IQR, interquartile range.

**Table 2 cancers-14-05530-t002:** Clinical characteristics of treated and not treated patients.

	No HM Tx at All		HM Tx at Least after COVID-19 dx		*p* Value	No HM Tx at All		HM Tx		*p* Value
	n	%	n	%		n	%	n	%	
Sex					0.031					0.021
Female	34	31.8%	96	44.2%		34	31.8%	152	44.3%	
Male	73	68.2%	121	55.8%		73	68.2%	191	55.7%	
Age, median (IQR) [absolute range]	68 (56–79) [24–92]		61 (47–72) [18–95]		<0.001	68 (56–79) [24–92]		64 (50–74) [18–95]		0.003
Comorbidities Before COVID-19					0.031					0.093
No comorbidities	37	34.6%	86	39.6%		37	34.6%	131	38.2%	
1 comorbidity	29	27.1%	71	32.7%		29	27.1%	111	32.4%	
2 comorbidities	28	26.2%	28	12.9%		28	26.2%	53	15.5%	
3 or more comorbidities	13	12.1%	32	14.7%		13	12.1%	48	14.0%	
Baseline malignancy					<0.001					<0.001
ALL	4	3.7%	19	8.8%		4	3.7%	28	8.2%	
AML	17	15.9%	62	28.6%		17	15.9%	112	32.7%	
Lymphoma	32	29.9%	82	37.8%		32	29.9%	124	36.2%	
CLD	26	24.3%	9	4.1%		26	24.3%	13	3.8%	
MM	12	11.2%	29	13.4%		12	11.2%	43	12.5%	
MDS/MPS	16	15.0%	16	7.4%		16	15.0%	23	6.7%	
Anti-SARS-CoV-2 vaccination	26	24.3%	38	17.5%	0.149	26	24.3%	64	18.7%	0.203
COVID-19 severity					0.019					0.092
Asymptomatic	20	18.7%	64	29.5%		20	18.7%	86	25.1%	
Mild infection	11	10.3%	27	12.4%		11	10.3%	47	13.7%	
Severe infection	39	36.4%	83	38.2%		39	36.4%	131	38.2%	
Critical infection	37	34.6%	43	19.8%		37	34.6%	79	23.0%	
COVID-19 diagnostic period					<0.001					0.001
March 2020–September 2020	27	25.2%	56	25.8%		27	25.2%	91	26.5%	
October 2020–February 2021	25	23.4%	99	45.6%		25	23.4%	142	41.4%	
March 2021–November 2021	24	22.4%	29	13.4%		24	22.4%	47	13.7%	
December 2021–February 2022	31	29.0%	33	15.2%		31	29.0%	63	18.4%	
Any COVID-19 treatment	46	43.0%	66	30.4%	0.025	46	43.0%	127	37.0%	0.268
Day-30 mortality	45	42.1%	33	15.2%	<0.001	45	42.1%	94	27.4%	0.004

ALL, acute lymphoid leukaemia; AML, acute myeloid leukaemia; CLD, chronic lymphoproliferative disorder; COVID-19, coronavirus disease 2019; Dx, diagnosis; HM, haematological malignancy; IQR, interquartile range; MDS, myelodysplastic syndrome; MM, multiple myeloma; MPS, myeloproliferative syndrome; SARS-CoV-2, severe acute respiratory syndrome coronavirus 2; Tx, treatment.

**Table 3 cancers-14-05530-t003:** Univariable and multivariable analysis for risk factors for mortality (whole cohort).

	Univariable				Multivariable			
	*p* Value	HR	95% CI		*p* Value	HR	95% CI	
			Lower	Upper			Lower	Upper
Sex								
Female	-	-	-	-				
Male	0.965	0.993	0.735	1.342				
Age	<0.001	1.035	1.024	1.046	<0.001	1.033	1.019	1.047
Baseline malignancy								
ALL	-	-	-	-	-	-	-	-
AML	0.015	2.632	1.210	5.726	0.080	2.307	0.906	5.873
Lymphoma	0.656	1.198	0.541	2.650	0.957	0.974	0.369	2.567
CLD	0.832	1.108	0.430	2.860	0.082	0.365	0.117	1.137
MM	0.085	2.090	0.904	4.833	0.941	1.040	0.374	2.889
MDS/MPS	0.245	1.704	0.694	4.180	0.752	0.838	0.282	2.495
COVID-19 infection severity								
Asymptomatic	-	-	-	-	-	-	-	-
Mild infection	0.919	1.036	0.525	2.045	0.862	1.068	0.509	2.239
Severe infection	0.080	1.545	0.949	2.515	0.026	1.804	1.074	3.032
Critical infection	<0.001	4.598	2.896	7.299	<0.001	5.523	3.328	9.166
Comorbidities at COVID-19 onset								
No comorbidities	-	-	-	-	-	-	-	-
1 comorbidity	0.033	1.527	1.034	2.256	0.618	1.113	0.732	1.692
2 comorbidities	<0.001	2.630	1.746	3.960	0.015	1.767	1.119	2.793
3 or more comorbidities	0.014	1.817	1.129	2.924	0.066	1.680	0.967	2.918
Season COVID-19 diagnosis								
March 2020–September 2020	-	-	-	-	-	-	-	-
October 2020–February 2021	0.003	0.603	0.430	0.845	0.699	1.084	0.720	1.634
March 2021–November 2021	0.060	0.639	0.400	1.019	0.402	1.280	0.719	2.279
December 2021–February 2022	<0.001	0.384	0.222	0.666	0.728	0.880	0.428	1.810
Malignancy treatment?								
No	-	-	-	-				
Before COVID-19 diagnosis	0.053	0.717	0.512	1.004				
After COVID-19 diagnosis	<0.001	0.251	0.154	0.408				
Neutrophils at COVID-19 onset								
≤500	-	-	-	-				
501–999	0.650	0.858	0.442	1.665				
≥1000	0.082	0.706	0.475	1.045				
Lymphocytes at COVID-19 onset								
≤200	-	-	-	-	-	-	-	-
201–499	0.058	0.582	0.333	1.018	0.210	0.683	0.377	1.239
≥500	0.005	0.553	0.366	0.836	0.024	0.573	0.352	0.931
Chemotherapy induced neutropenia								
No	-	-	-	-	-	-	-	-
Yes	0.071	0.722	0.506	1.029	0.239	1.285	0.846	1.950
No treatment administered	0.011	1.602	1.112	2.308	<0.001	3.449	2.224	5.349
Secondary infection after COVID-19	0.125	0.740	0.503	1.087				
COVID-19 treatment								
No specific treatment reported	-	-	-	-	-	-	-	-
Antivirals + monoclonal antibodies ± corticosteroids ± plasma	0.213	0.286	0.040	2.049	0.105	0.190	0.025	1.414
Antivirals ± corticosteroids ± plasma	0.793	1.064	0.668	1.695	0.395	0.798	0.476	1.341
Corticosteroids	0.791	0.944	0.617	1.444	0.033	0.605	0.381	0.961
Monoclonal antibodies ± plasma ± corticosteroids	0.026	0.360	0.147	0.882	0.081	0.442	0.177	1.107
Plasma ± corticosteroids	0.905	1.063	0.392	2.880	0.269	0.557	0.198	1.571

ALL, acute lymphoid leukaemia; AML, acute myeloid leukaemia; CI, confidence interval; CLD, chronic lymphoproliferative disorder; COVID-19, coronavirus disease 2019; HR, hazard ratio; MDS, myelodysplastic syndrome; MM, multiple myeloma; MPS, myeloproliferative syndrome asymptomatic: no clinical signs or symptoms; mild: non-pneumonia and mild pneumonia; severe: dyspnoea, respiratory frequency ≥ 30 breaths per min, SpO_2_ ≤ 93%, PaO_2_/FiO_2_ < 300, or lung infiltrates > 50%); critical: patients admitted in intensive care for respiratory failure, septic shock, or multiple organ dysfunction or failure.

## Data Availability

The datasets generated during and/or analysed during the current study are available from the corresponding author on reasonable request.

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
