# Peer review of "Simultaneous Onset of Haematological Malignancy and COVID: An Epicovideha Survey"

_cancers, 2022, doi:10.3390/cancers14225530_

Round 1
Reviewer 1 Report
Although at the end of the introduction authors list 5 aims that this study is going to address, I am still a little confused about the general idea/ specific aims or proposed hypotheses (if any) in terms of the study design. Just looking at the abstract and introduction, I could not tell if the study was trying to address how COVID-19 infection affects HM treatment or how HM treatment affects the prognosis/outcome of the simultaneous COVID-19 infection. If COVID-19 infection status and its corresponding severity are evaluated for the treatment response rate of certain hematological malignancies, the data from HM cohort without COVID-19 infection should definitely be included. And for claims like "HM treatment is needed for a favorable outcome in patients with simultaneous diagnosis of COVID-19 and HM" is not an appropriate/specific conclusion because it doesn't need the study to tell and any people with common sense know it.
Author Response
See attached answer to reviewers

Reviewer 2 Report
The paper investigated the impact of prompt versus delayed chemotherapy in a series of heterogeneous patients with simultaneous diagnosis of hematologic disease and COVID infection.
The paper contains many interesting informations despite the limitations already underlined by authors.
The are just few questions to be answered
1. In table 1, listing patients characteristics there is a small proportion of them with lymphopenia. Was lymphopenia associate with higher infection probability or death? Did the authors consider hypogammaglobulinemia?
2. The difference in 30 day survival probability is probably due to the high proportion of AML/mds patients. This group should be separately analyzed.
3. In lymphoproliferative disorders risk of death seems lower than in myeloproliferative ones, despite studies of sieroconversion after vaccines demonstrate lower response rate in lymphoproliferative than in myeloproliferative diseases. Please add a comment.
4. If neutropenia is not a risk factor for death in AML, how did they justify the high mortality in this subgroup?
5. They reported very low response rate to treatment. How did they explain this fact?
Author Response
see attached answer to reviewers

Round 2
Reviewer 1 Report
The authors' efforts to address reviewers' comments are acknowledged.
Reviewer 2 Report
I read the revised paper. No further questions